:ᐰ: PLOS | ONE

# Pseudolesion in the right parafissural liver parenchyma on CT: The base is found in embryology and collagen content

Willemijn M. Klein[1]*, Lianne J. P. Sonnemans[1], Sabine Franckenberg[2¤a], Barbara Fliss[2], Dominic Gascho[2], Mathias Prokop[1], Wouter H. Lamers[3], Jill P. J. M. Hikspoors[3], Michael J. Thali[2], Patricia M. Flach[2¤b]

1 Department of Radiology and Nuclear Medicine, Radboud University Medical Center, Nijmegen, the Netherlands, 2 Department of Forensic Medicine and Imaging, Institute of Forensic Medicine, University of Zürich, Zürich, Switzerland, 3 Department of Anatomy and Embryology, Maastricht University, Maastricht, the Netherlands

¤a Current address: Institute for Diagnostic and Interventional Radiology, University Hospital Zurich, University of Zurich, Zürich, Switzerland
¤b Current address: Clinic for Radiology and Nuclear Medicine, Cantonal Hospital St. Gallen, St. Gallen, Switzerland
* Willemijn.Klein@radboudumc.nl

**Data Availability Statement:** Anonymized and compact data are available in the Supporting Information files.

## Abstract

### Background

Computed tomography (CT) images of livers may show a hypo-attenuated structure alongside the falciform ligament, which can be a focal fatty pseudolesion and can mimic a malignancy. The preferred location is on the right parafissural site, ventral in segment IVa/b. The etiology is not clear, nor is it known how the histology of this location develops. These are evaluated in this study.

### Methods

40 adult cadavers with autopsy and / or postmortem CT in a university hospital and a forensic center were included. Liver biopsies were taken at the left side of the falciform ligament as control, and at the right side as the possible precursor of a pseudolesion; these were examined for collagen and fat content. Cadavers with steatotic (>5% fat) or fibrotic (>2% collagen) control samples were excluded.

### Results

Significantly more collagen was present in the right parafissural liver parenchyma: median 0.68% (IQR: 0.32–1.17%), compared to the left side 0.48% (IQR: 0.21–0.75%) (p 0.008), with equal fat content and CT attenuation values. The etiophysiology goes back to the demise of the umbilical venes in the early embryonic and neonatal period.

**Funding:** None of the authors have anything to declare. The authors received no specific funding for this work.

**Competing interests:** The authors have declared that no competing interests exist.

**Abbreviations:** BMI, body mass index; EVG, elastic Van Gieson; IQR, Interquartile range; PMCT, postmortem computed tomography; RGB, red-green-blue; SD, standard deviation.

## Conclusions

The right parafissural area contains more collagen and an equal amount of fat compared to the control left side. This supports the hypothesis of delayed, 'third' inflow: the postnatal change in blood supply from umbilical to portal leaves the downstream parafissural area hypoperfused leading to hypoxia which in turn results in collagen accumulation and the persistence of paraumbilical veins of Sappey.

## Introduction

Contrast-enhanced computed tomography (CT) scans of the liver demonstrate a pseudolesion in 20% of patients in liver segment IV, directly adjacent to the right side of the falciform ligament [1,2]. This pseudolesion presents as a focal hypo-attenuated area, usually visible in the portovenous phase after administration of an intravenous contrast agent. Pseudolesions may be difficult to differentiate from malignancy such as a metastasis or hepatocellular carcinoma. Although pseudolesions can be found in other areas of the liver, such as alongside the gall bladder, the typical location is to the right of the falciform ligament.

There have been several studies on the radiological appearance of pseudolesions. Typically, the pseudolesion is hypodense in the portovenous phase, which could result from preferred late ('third') inflow via the paraumbilical veins in this area, instead of inflow via the hepatic artery and portal vein [3,4]. A pseudolesion is also labeled as a "focal fatty change", as on magnetic resonance imaging (MRI) this area can contain more fat than the surrounding normal tissue [1]. The single study on the histology of the liver pseudolesion showed a slightly higher collagen content on the right side of the falciform ligament, compared to the left [5]. However, the genesis, (patho)physiology and histology of this right parafissural area are still unclear. Clinicians and radiologists can be more confident in their (benign or malignant) diagnosis and avoid unnecessary biopsies if the etiology of the pseudolesion were understood.

The authors hypothesize that the histology of the right parafissural location of a healthy liver differs from the contralateral side, even when a pseudolesion is not visible on the CT images. This area is typically predisposed to develop a pseudolesion, the origin of which may be found in the normal development of the liver and histology. The purpose of this study is to evaluate why this location is predisposed to becoming a pseudolesion.

## Materials and methods

### Inclusion

This study was conducted in a university hospital and in a forensic institute. Cases included in the hospital setting were adult cadavers in the period December 2013 to February 2014 on whom a body autopsy was performed. Part of the results were published previously as a single center study [5]. To increase the study population in number and variety, inclusion criteria were extended to a forensic institute from October 2014 until December 2014. Inclusion was non-consecutive and subject to the availability of personnel and time. All authors had access to the study data and reviewed and approved the final manuscript. All forensic cases had died with questionable or obvious non-natural causes and were therefore subject to forensic investigation, according to the law. The cadavers were included if they were over 18 years of age and had undergone a full body postmortem CT scan. All cadavers were 'fresh', as they had died

within 1 to 3 days prior to scanning and autopsy, were kept in a cooled room and showed early but not late signs of decomposition.

As the subjects of study are deceased, informed consent was obtained from the relatives in case of clinical cadavers, and from the prosecutor in case of forensic cadavers. In the clinical cases, permission for clinical autopsy, including liver biopsy, was given by the bereaved family. Ethical consent for this postmortem study was waived by the Institutional Review Board of the study hospital (Radboud university medical center, Nijmegen, the Netherlands), according to the national law and local guidelines and regulations (2012–280). In the forensic cases, permission for CT guided liver biopsy was given by the prosecutor (in cases where no autopsy was performed), or by the pathologist and radiologist (in cases where autopsy was performed, and liver biopsies were unlikely to obstruct the proper forensic autopsy), which was according to the national law, local guidelines and regulations. Ethical approval was waived by the responsible ethics committee of the Canton of Zurich, Switzerland (waiver number 2015–0686). All methods were carried out in accordance with relevant guidelines and regulations.

The cause of death was based on all available data, including the postmortem CT scan and the autopsy. In case of a combination of contributing diseases, the direct cause was assigned as cause of death.

## CT scanning parameters

CT scans were performed on a 64 slice scanner. Clinical postmortem CT (PMCT) could be performed on either Siemens Somatom Sensation 16 or Siemens Sensation 64 (Siemens Healthcare, Forchheim Germany) with a detector collimation of 1 mm, reconstruction interval of 0.8 mm and 120 kV, with a tube current of 400 mA and 1 s rotation time. All subjects scanned at the forensic institute underwent the following full body PMCT scan protocol using a standard medical CT scanner (SOMATOM Flash Definition, Siemens, Forchheim, Germany). Automated dose modulation (CARE Dose4DTM, Siemens, Forchheim, Germany) with 400 reference mAs was used for all scans with a tube voltage of 120 kV. The pitch factor was 0.35 and the rotation time 0.5 s. With slice thickness of 1 mm and increment of 0.6, there were reconstructions in a soft kernel (B30) and abdominal window, in a hard kernel (B60) and pulmonary window; with slice thickness of 5 mm and increment of 3, there were reconstructions in a soft kernel (B30) and abdominal window [6].

## CT attenuation

The attenuation (Hounsfield units, HU) of the liver parenchyma on the right and left side of the falciform ligament was measured on 5 mm reconstruction slices with a region of interest (ROI) of 0.40 cm2 using PACS software (Siemens, Erlangen, Germany) by one observer (either LS or WK).

## Histology

In the forensic cadavers, the image-guided biopsies were obtained of the right side of the falciform ligament, corresponding to the border area between segments IVa and IVb. Control biopsies were taken on the left side of the falciform ligament, corresponding to the border area between segment II and III (Fig 1). A 22mm 14Gauge biopsy needle was used by placement of a co-axial needle and using a semi-automated biopsy gun (BARD biopsy, Tempe, Arizona, USA). Biopsies were taken three times on each side. In the clinical cadavers, the tissue samples were taken during autopsy, on the right and left side of the falciform ligament, at the level of the left portal vein [5].

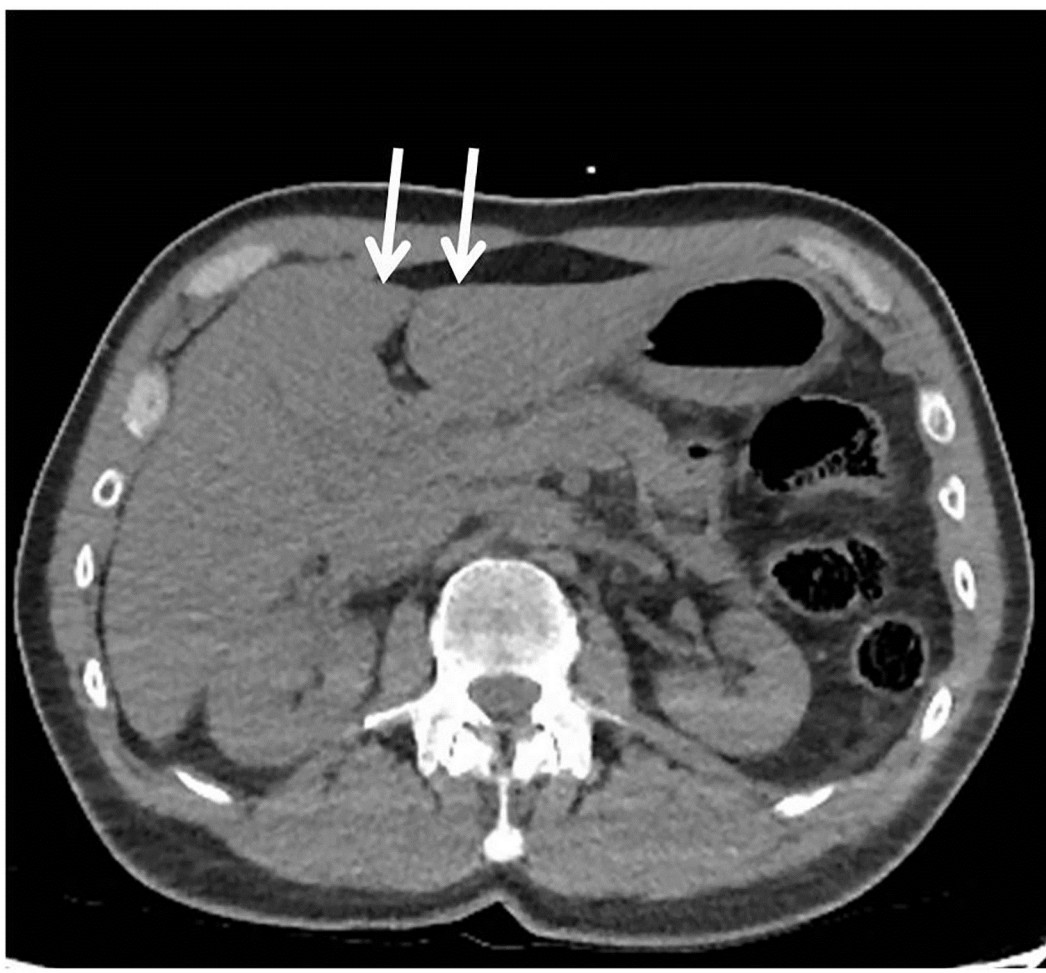

**Fig 1. Biopsy locations.** Postmortem CT slice through the liver. The arrows indicate the site of the biopsies on the left and right site of the falciform ligament.

The liver tissue specimens were fixed in a 10% formalin solution, processed in a histoprocessor and embedded in paraffin. From every specimen, a 4 μm section was stained with Elastic Van Gieson (EVG). Digital images were recorded by an AxioPhot microscope and an AxioCam MRC (Zeiss, Jena, Germany) using a 10x objective lens (N.A. = 0.30) with a resulting specimen level pixel size of 1.06 μm. Using digital image analysis by KS400 software (Zeiss, Jena, Germany), a variable number of field-of-views (depending on the size of the section, but with a maximum of twenty) were recorded and measured quantitatively for the amount of intraparenchymal collagen and fat. All images were corrected for unequal illumination using an empty field image. Based on the red-green-blue (RGB) camera signal, the pink fuchsin dye of the EVG-staining was automatically recognized as collagen and small circular empty regions were automatically recognized as being fat vacuoles. For every biopsy the mean surface-percentage collagen and fat present in a field-of-view was calculated. Large vessels were placed outside the field-of-view, to prevent them from being measured, as this study intends to measure histological differences in parenchymal tissue only. The observer (LS) was blinded to the site of origin of the specimens and the outcome of the corresponding CT images.

## Statistical analysis

The authors intended to study liver parenchymal histology at the site that is predisposed of becoming a pseudolesion in individuals without significant liver disease. For this reason, patients with liver fibrosis or steatosis in the control tissue of the left side were excluded from the statistical analysis. Fibrosis was defined as >2% collagen and steatosis as >5% fat content [7–9].

Wilcoxon's signed rank tests were used to test for differences of CT attenuation and histology of the collagen and fat content on the left and right side of the falciform ligament.

Significance level was established at $p < 0.05$. The software used for analysis was IBM SPSS (IBM SPSS Statistics for Windows, Version 25.0. Armonk, NY, United States of America).

## Etiophysiology

Including the available literature and the results of the current study, we aim to reconsider the etiology and development of liver parenchyma, especially the area that is predisposed to becoming a pseudolesion.

## Results

Liver biopsies of 52 cadavers were collected (32 men and 20 women, mean age 56.2 years, SD 15.2). Two cadavers were excluded because the histological results of the right side of one cadaver and both sides of another cadaver were inadvertently not recorded. As a result 50 pairs of biopsies were available for comparison. After exclusion of cadavers with >2% fibrosis or >5% steatosis content in the control biopsy on the left side, this resulted in the inclusion of 40 cadavers, 23 from the forensic institute and 17 from the university hospital. Patient characteristics are shown in Table 1.

**Table 1. Basic characteristics.** Basic characteristics of the 40 included cadavers (median, interquartile range IQR).

|  | Mean (SD) |
|---|---|
| Sex M:F (%) | 23:17 (58:43%) |
| Age | 59.6 years (48.0–67.8) |
| Body mass index (n = 36) | 27.7 (23.1–31.4) |
| Liver weight (n = 36) | 1699 g (1383–2273) |
| Cause of death * |  |
| Natural:<br>• Cardiovascular disease<br>• Cardiopulmonary disease<br>• Cancer and sequels<br>• Infectious disease | 13 (33%)<br>6 (17%)<br>4 (11%)<br>8 (22%) |
| Non-natural:<br>• High energy impact<br>• Violence<br>• Drug intoxication<br>• Asphyxia | 1 (2%)<br>1 (2%)<br>5 (13%)<br>2 (6%) |

*Cardiovascular: including myocardial infarction, aortic dissection, arrhythmia and hemorrhage. Cardiopulmonary: including pulmonary embolism and cardiopulmonary insufficiency or decompensation. Cancer: including sequels of cancerous disease. Infectious disease: including sepsis and infectious diseases of the heart and lungs. High impact trauma: fall from height. Violence: lethal injuries by bullets. Drug intoxication: both acute and chronic. Asphyxia: including carbon monoxide intoxication.

## CT attenuation

A postmortem CT was performed in 24 of 40 cases. The attenuation value in the livers was not different with 49 HU (IQR: 39–60) on the right side and 48 HU (IQR: 40–61) on the left side (p 0.16). There were two cases with a visible pseudolesion on the unenhanced postmortem CT.

## Histology

In the 40 non-fibrotic and non-steatotic livers, the median collagen content at the right and left side of the falciform ligament were 0.68% (IQR: 0.32–1.17%) and 0.48% (IQR: 0.21–0.75) respectively, which was a statistically significant difference (p 0.008). The median fat percentage at the right and left side of the falciform ligament was 0.45% (IQR: 0.16–0.94%) and 0.32% (IQR: 0.14–0.99%), respectively, which was not different (p 0.91).

## A pseudolesion on postmortem CT

In two cases a typical hypoattenuated area on the right side of the falciform ligament was visible on the non contrast-enhanced postmortem CT (Fig 2). Case A was a 54 year old woman

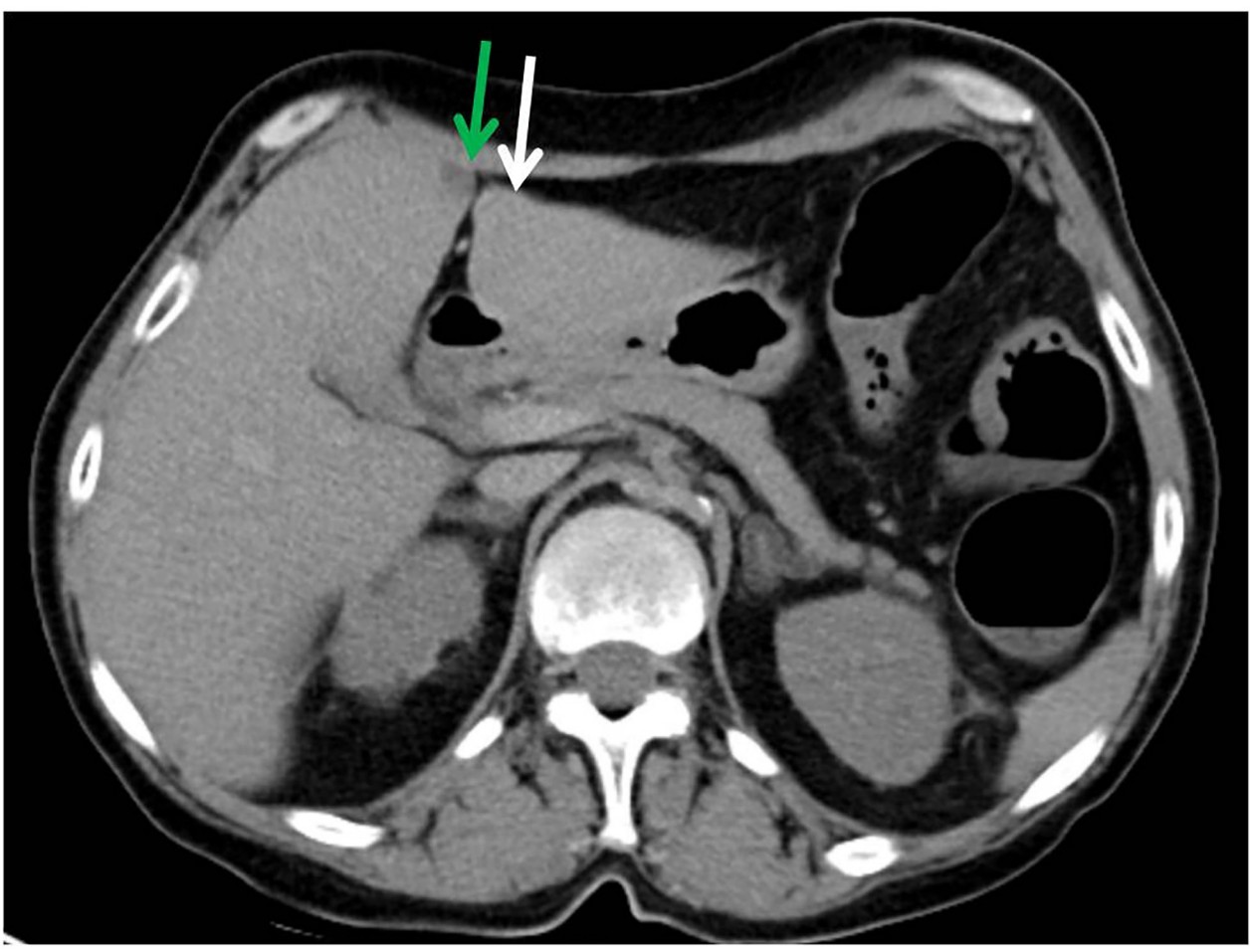

**Fig 2. Case A with a pseudolesion.** Axial slice of the non contrast-enhanced postmortem CT scan, through the liver of a 54 year old woman who died of natural cardiac cause. CT attenuation values and collagen and fat content on the right parafissural side (green arrow) were 39 HU, 1.17% and 0.51% respectively, compared to 64 HU, 0.68% and 0.53% on the left side (white arrow).

**Table 2. Cases with a pseudolesion.** Cases with a pseudolesion on the non-contrast-enhanced postmortem CT on the right side of the falciform ligament.

| | Case A | Case B |
|---|---|---|
| CT attenuation (HU) | | |
| Right | 39 | 16 |
| Left | 64 | 47 |
| Collagen content (%) | | |
| Right | 1.17 | 0.48 |
| Left | 0.68 | 0.14 |
| Fat content (%) | | |
| Right | 0.51 | 1.81 |
| Left | 0.53 | 1.54 |

who died of natural cardiac causes. The CT attenuation value was 39 HU on the right and 64 HU on the left parafissural side. (Table 2) Her liver weighed 1720 g (normal values 603–1767 g) [10]. Histology of case A showed higher collagen content on the right side of the falciform ligament (1.17% versus 0.68%) and equal fat content (0.51% versus 0.53%) compared to the left side. Table 2

Case B was a 32 year old woman who died of suicidal drug intoxication. The CT attenuation value was 16 HU on the right and 47 HU on the left parafissural side. Her liver weighed a heavy 2450 g. Histology of the liver parenchyma showed a higher collagen (0.48% versus 0.14%) and slightly higher fat content (1.81% versus 1.54%) on the right compared to the left parafissural side. Table 2 A paraumbilical vein was identified just ventral of the pseudolesion. (Fig 3).

## Etiophysiology

Combining the knowledge of the available literature and our histological studies on the preferred location of the liver pseudolesion gives direction to propose the etiobiology of the liver pseudolesion. It has been suggested that the pseudolesion is a perfusion defect and related to aberrant venous supply or 'third' inflow via the paraumbilical veins [1,2,4,11–14]. CT and MRI studies have suggested that pseudolesions are focal fatty deposition or, on the opposite, sparing. Our current and previous studies are the only histological studies on this subject [5]. We demonstrated a significantly higher amount of collagen on the right parafissural side in healthy, non-fibrotic and non-steatotic livers. This histological right-left difference in non-fibrotic and non-steatotic livers, not selected for pseudolesions, is suggestive for a common developmental physiology. A recent embryological study is confirmative in this direction [15]. Hikspoors et al. studied the embryologic development of liver veins using histological sections of human embryos aged 5 to 10 weeks, reconstructed to three-dimensional models. The ventromedial lobe of the embryonic liver (future segments IV-VI) is initially supplied by both umbilical veins (Fig 4) [16]. The right umbilical vein normally involutes at 5 weeks of development (Carnegie stage 15), whereas the left one involutes after birth and becomes the round ligament, positioned within the falciform ligament. The umbilical veins also drain the ventral body wall, in agreement with their evolutionary origin as the abdominal vein. Small collaterals of the umbilical vein that persist are known as paraumbilical or Sappey's veins [3]. (Fig 5) It is not known why Sappey's veins can persist, but we posit that after birth, when the left umbilical vein regresses and the left portal vein takes over as main blood source for the left-sided ventromedial lobe, not all parts of this downstream area become well perfused via the longer and more indirect route of the left portal vein. This hypoperfusion would allow the persistence of

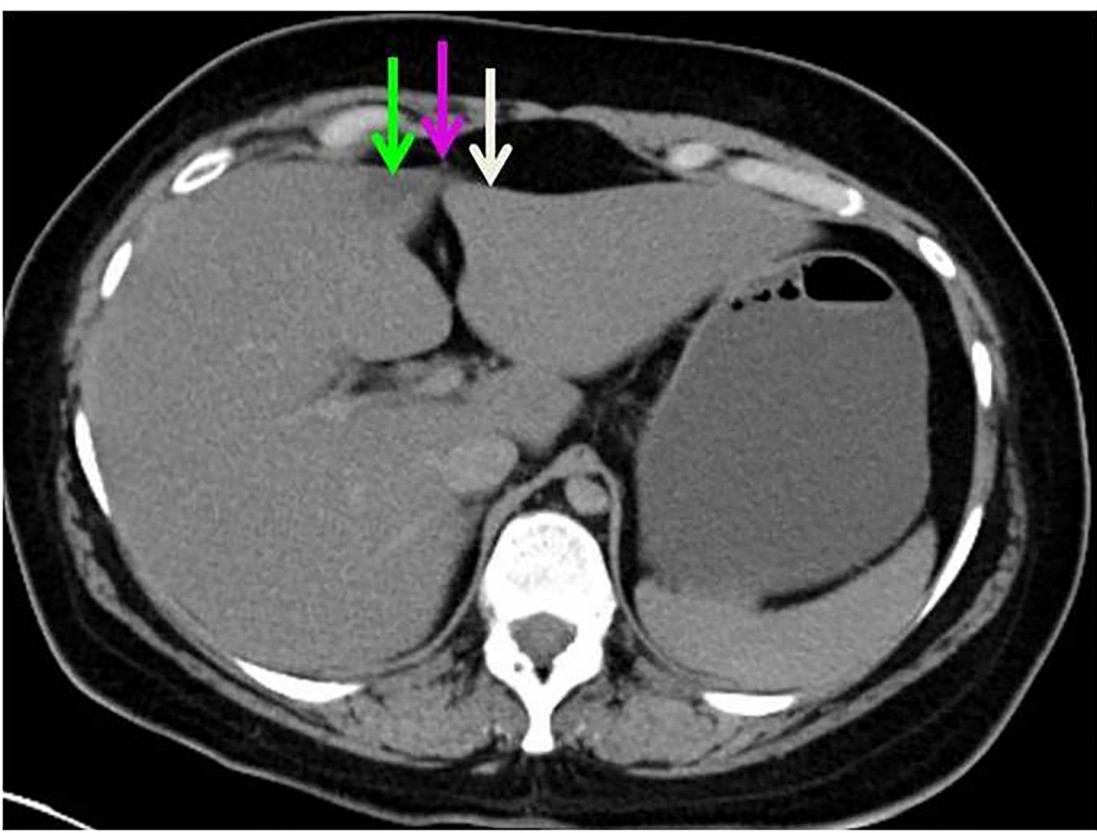

**Fig 3. Case B with a pseudolesion.** Axial slice of the non-contrast-enhanced postmortem CT scan, through the liver of a deceased 32 year old female (case B) who died of drug intoxication. The right parafissural side (green arrow) shows a pseudolesion with low attenuation compared to the left parafissural side (white arrow) The purple arrow indicates a paraumbilical vein.

the paraumbilical veins in the abdominal wall as source of afferent blood. The different perfusion causes late enhancement, which is typically visible in the portal phase of contrast-enhanced CT imaging, because of delayed 'third' inflow of contrast via the ventral body wall (paraumbilical) veins. Apparently, this source is qualitatively suboptimal, as the abnormally perfused area accumulates collagen fibers, which is a known consequence of hypoxia [17,18].

## Discussion

In this study, we demonstrated that the right parafissural parenchyma of non-fibrotic and non-steatotic liver, which is the location that is prone to developing a hypoattenuated pseudolesion, contains more collagen than the left side, with equal amounts of fat. Our hypothesis is that this pseudolesion results from insufficient perfusion after postnatal regression of the umbilical vein. This hypoperfusion leads to local deposition of collagen fibers, as well as to the persistence of the paraumbilical veins of Sappey.

Our previous study on the histology on the parafissural liver parenchyma indicated a higher amount of collagen on the right side. However, the low inclusion rates and few (postmortem) CT scans did not allow us to test the hypothesis properly [5]. We therefore extended the study using improved methods including a postmortem CT scan for every case and CT guided biopsies. We were now able to demonstrate that the right side of the parafissural liver parenchyma contains significantly more collagen than the left side, in non-fibrotic and non-steatotic livers,

Fig 4A

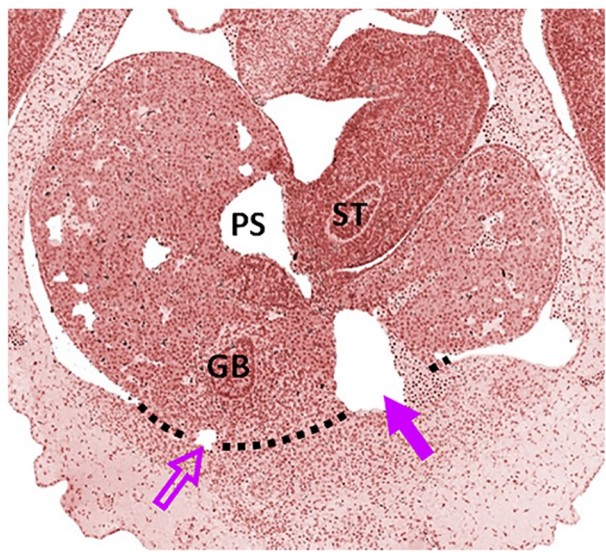

Fig 4B

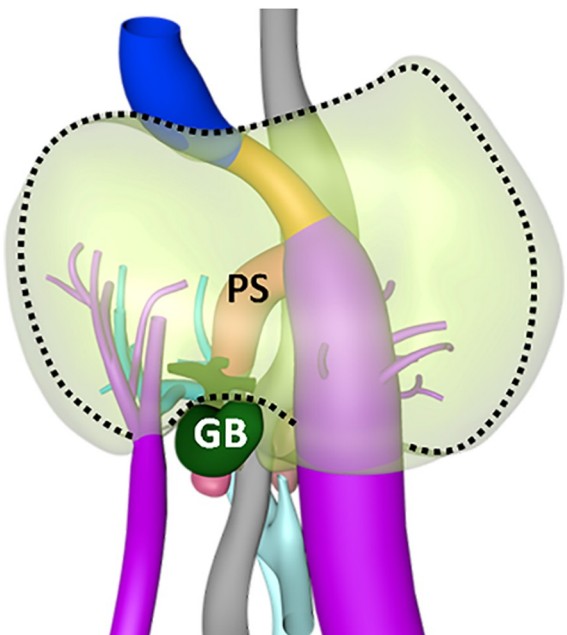

**Fig 4. Embryonic liver.** Histological section (Fig 4A) and 3D model (Fig 4B) of the human embryonic liver aged 34 days. The open purple arrow indicates the involuting right umbilical vein; the solid purple arrow the left umbilical vein. The black dotted line marks the connection of the ventral side of the liver to the ventral body wall, which hardly changes over time and becomes a thin structure (falciform ligament) as the volume of the liver increases about 40-fold between 5 and 8 weeks of development. GB—Gallbladder, ST—Stomach, PS—Portal sinus. Color code: blue—Inferior caval vein, light blue—Portal vein, grey—Gastro-intestinal tract.

Fig 5A

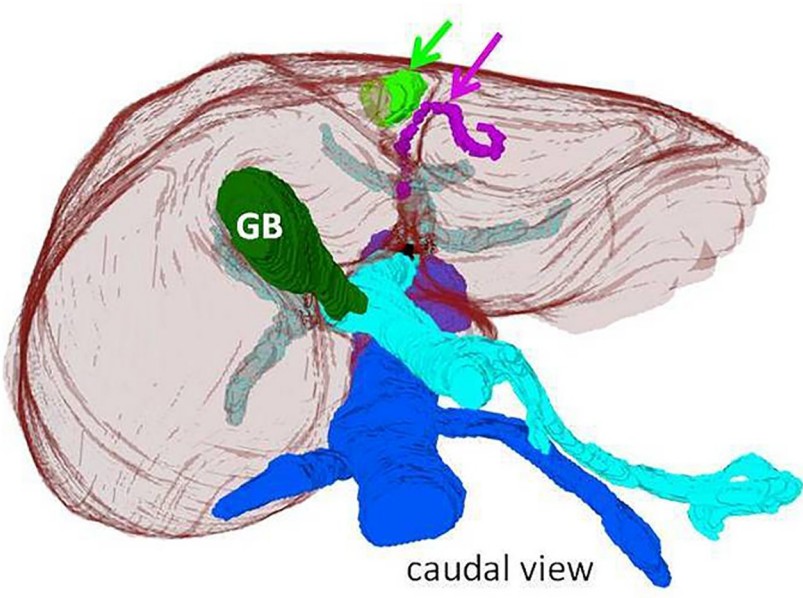

caudal view

Fig 5B

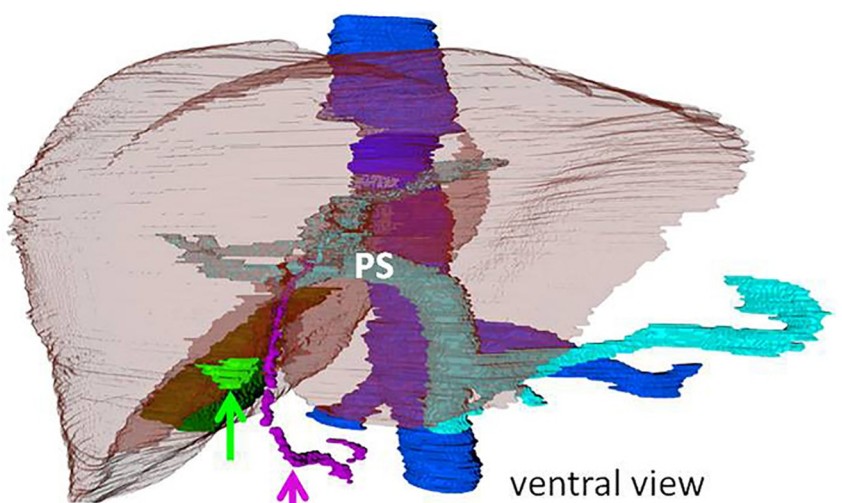

ventral view

**Fig 5. Adult liver.** Caudal (Fig 5A) and ventral (Fig 5B) view on 3D reconstructions of a contrast-enhanced CT of an adult liver (in transparent brown). The pseudolesion on the right parafissural side is indicated in green (green arrow). The purple arrow indicates a paraumbilical vein of Sappey, which lies in the falciform ligament. GB—Gallbladder, in dark green. PS—Portal sinus, in light blue. Inferior caval vein, in dark blue.

in a population not selected for pseudolesions. This result supports the hypothesis that a disadvantageous hypoxic blood supply leads to local collagen deposition and maintenance of the collateral paraumbilical veins [1].

Our study has some limitations. All inclusions were postmortem. Normal postmortem changes could have influenced the attenuation values of the liver on CT. It is known that the liver decreases in size after death probably through the sedimentation of blood to the depending part of the body (livor mortis) [19]. Therefore it is likely that the attenuation values of the liver change after death, as is also seen in the postmortem spleen [20]. This could explain the slightly lower liver attenuation values in our study population (median of 49 HU) compared to healthy living persons (around 55 HU) [21]. However, it is not likely that the collagen or fat content changes shortly after death, nor that the left and right side of the falciform ligament would have a different pattern of postmortem changes. Further, due to postmortem circumstances it was not possible to scan with intravenous contrast in a portovenous phase, which is the most likely moment to visualize pseudolesions. Therefore we cannot confirm that more cases in our study group would have shown a liver pseudolesion on enhanced CT, besides the two that were visible on the unenhanced CT images. Also, due to the postmortem collapsing of the veins we were not able to study the presence of paraumbilical veins of Sappey. Further research would be of great interest, for instance performing liver biopsies in laparoscopy patients with a pseudolesion noted on CT images in portovenous enhancement phase, if consented for.

The cases were not typically normal healthy persons, as they all had deceased by either a natural lethal disease, non-natural traumatic or a forensic cause. Several of them had a history of alcohol and drug abuse, therefore it is unlikely that all had normal healthy livers. Still, the choice for a postmortem study was necessary to obtain histology in all cases. Fibrotic and steatotic cases were excluded to diminish the effect of liver disease as a possible confounder.

We conclude that the non-fibrotic and non-steatotic liver has a significantly higher content of collagen on the right parafissural side. This is the typical location for a pseudolesion. We suggest the embryologic and neonatal vascular development of the liver to be the physiological basis. This is in concordance with other findings correlated to pseudolesions, such as late enhancement, third inflow and metabolic (fibrotic) alterations.

## Supporting information

**S1 Data.**
(XLSX)

**S1 File.**
(DOCX)

## Author Contributions

**Conceptualization:** Willemijn M. Klein, Mathias Prokop, Michael J. Thali, Patricia M. Flach.

**Data curation:** Willemijn M. Klein, Lianne J. P. Sonnemans, Sabine Franckenberg, Barbara Fliss, Dominic Gascho.

**Formal analysis:** Willemijn M. Klein, Lianne J. P. Sonnemans, Mathias Prokop.

**Funding acquisition:** Willemijn M. Klein, Mathias Prokop, Michael J. Thali, Patricia M. Flach.

**Investigation:** Willemijn M. Klein, Lianne J. P. Sonnemans, Barbara Fliss, Dominic Gascho, Wouter H. Lamers, Jill P. J. M. Hikspoors, Patricia M. Flach.

**Methodology:** Willemijn M. Klein, Lianne J. P. Sonnemans, Mathias Prokop, Wouter H. Lamers, Jill P. J. M. Hikspoors, Patricia M. Flach.

**Project administration:** Willemijn M. Klein, Lianne J. P. Sonnemans, Sabine Franckenberg, Barbara Fliss, Dominic Gascho, Patricia M. Flach.

**Resources:** Willemijn M. Klein, Sabine Franckenberg, Barbara Fliss, Dominic Gascho, Mathias Prokop, Michael J. Thali, Patricia M. Flach.

**Supervision:** Willemijn M. Klein, Mathias Prokop, Wouter H. Lamers, Michael J. Thali, Patricia M. Flach.

**Validation:** Willemijn M. Klein, Lianne J. P. Sonnemans, Mathias Prokop, Jill P. J. M. Hikspoors, Patricia M. Flach.

**Visualization:** Willemijn M. Klein.

**Writing – original draft:** Willemijn M. Klein, Lianne J. P. Sonnemans, Mathias Prokop, Wouter H. Lamers, Jill P. J. M. Hikspoors, Patricia M. Flach.

**Writing – review & editing:** Willemijn M. Klein, Lianne J. P. Sonnemans, Sabine Franckenberg, Barbara Fliss, Dominic Gascho, Mathias Prokop, Wouter H. Lamers, Jill P. J. M. Hikspoors, Michael J. Thali, Patricia M. Flach.

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
