## [Decision Letter · Decision Letter 0]

11 Nov 2019

PONE-D-19-22279

Pseudolesion in the right parafissural liver parenchyma on CT: the base is found in embryology and collagen content.

PLOS ONE

Dear Dr. Klein,

Thank you for submitting your manuscript to PLOS ONE. After careful consideration, we feel that it has merit but does not fully meet PLOS ONE’s publication criteria as it currently stands. Therefore, we invite you to submit a revised version of the manuscript that addresses the points raised during the review process.

Please edit the discussion for readability and clarity.  No additional data is required.

We would appreciate receiving your revised manuscript by Dec 26 2019 11:59PM. To enhance the reproducibility of your results, we recommend that if applicable you deposit your laboratory protocols in protocols.io, where a protocol can be assigned its own identifier (DOI) such that it can be cited independently in the future. For instructions see: http://journals.plos.org/plosone/s/submission-guidelines#loc-laboratory-protocols

We look forward to receiving your revised manuscript.

Kind regards,

Leonidas G Koniaris, MD

Academic Editor

PLOS ONE

Journal Requirements:

Additional Editor Comments:

This is an interesting study that will be of interest. Please edit the discussion for clarity and readability.

Reviewers' comments:

Reviewer's Responses to Questions

**Comments to the Author**

1. Is the manuscript technically sound, and do the data support the conclusions?

Reviewer #1: Yes

2. Has the statistical analysis been performed appropriately and rigorously? 

Reviewer #1: Yes

3. Have the authors made all data underlying the findings in their manuscript fully available?

Reviewer #1: Yes

4. Is the manuscript presented in an intelligible fashion and written in standard English?

Reviewer #1: Yes

5. Review Comments to the Author

Reviewer #1: Authors are commended for pursuing research to explain a common finding on liver imaging - 'focal fatty sparing' adjacent to the falciform ligament

Suggest:

1. This may be corroborated by performing liver biopsies at laparotomy in patients found to have the finding on imaging: obviously under research consent

2. The discussion can be made more succinct. The embryological images do not add much to the discussion.

6. PLOS authors have the option to publish the peer review history of their article (what does this mean?). If published, this will include your full peer review and any attached files.

Reviewer #1: No

---

## [Author Response · Author response to Decision Letter 0]

14 Dec 2019

Dear editor,

Thank you for your thorough review of our manuscript and thank you for welcoming a revised version. 

You have advised us to edit the discussion for readability and clarity, without requirement for additional data. The reviewer advised us to make the discussion more succinct. We agree that this would help to improve the manuscript and we have applied this. In the track& trace version you can see the adjustments made in the discussion, that make the discussion more readable and succinct. 

Further you have advised financial updates (that we have none) and journal style requirements (that we have all applied according to the Plos ONE style). Also I have enclosed a FAIR dataset that is fully anonymized that can be published with the manuscript. Numbers have been 

Journal requirements 1: the title, authors and main body have been adjusted.

Journal requirements 2 and 3: A FAIR dataset is now added. We have supplied a minimal dataset with rounded numbers (age) to diminish any recognizability in the context of privacy.

Reviewer: 

Suggestion 1. 

The reviewer argues that the embryological images do not add much to the discussion. We disagree. It is one of the main messages of this manuscript that the cause of a supposedly abnormal anatomy can be found in embryology. We find it important to show the embryological evaluation of the umbilical veins to support our hypothesis. Therefore we wish to leave the images as these are, illustrating the paragraph Etiophysiology in the Results. 

Suggestion 2. 

The reviewer gives a suggestion how to further study the liver focal fatty sparing, namely by biopsy during laparotomy. We have added this to the discussion and thank the reviewer for his valuable addition. 

We have adjusted the discussion to make it more succinct. We thank the reviewer for his thorough review of our manuscript and for helping us to improve it. 

We look forward to your decision concerning our manuscript. Thank you for your important contribution to this study, concerning the liver from embryology to postmortem, in the matter of the parafissural pseudolesion. Our results will be helpful to the everyday work of radiologists and hepatologists all over the world.

---

## [Editor Report · Decision Letter 1]

3 Jan 2020

Pseudolesion in the right parafissural liver parenchyma on CT: the base is found in embryology and collagen content.

PONE-D-19-22279R1

Dear Dr. Klein,

We are pleased to inform you that your manuscript has been judged scientifically suitable for publication and will be formally accepted for publication once it complies with all outstanding technical requirements.

With kind regards,

Leonidas G Koniaris, MD

Academic Editor

PLOS ONE
---

## [Editor Report · Acceptance letter]

7 Jan 2020

PONE-D-19-22279R1 

Pseudolesion in the right parafissural liver parenchyma on CT: the base is found in embryology and collagen content. 

Dear Dr. Klein:

I am pleased to inform you that your manuscript has been deemed suitable for publication in PLOS ONE. Congratulations! Your manuscript is now with our production department. 

With kind regards,

on behalf of

Dr. Leonidas G Koniaris 

Academic Editor

PLOS ONE